# The Effects of a Preconception Lifestyle Intervention on Childhood Cardiometabolic Health—Follow-Up of a Randomized Controlled Trial

**DOI:** 10.3390/cells11010041

**Published:** 2021-12-24

**Authors:** Stijn Mintjens, Mireille N. M. van Poppel, Henk Groen, Annemieke Hoek, Ben Willem Mol, Rebecca C. Painter, Reinoud J. B. J. Gemke, Tessa J. Roseboom

**Affiliations:** 1Department of Pediatrics, Emma Children’s Hospital, Amsterdam UMC, 1105 AZ Amsterdam, The Netherlands; rjbj.gemke@amsterdamumc.nl; 2Department of Obstetrics and Gynecology, Amsterdam UMC, 1105 AZ Amsterdam, The Netherlands; r.c.painter@amsterdamumc.nl (R.C.P.); t.j.roseboom@amc.uva.nl (T.J.R.); 3Department of Pediatrics, NYC Health & Hospitals/Lincoln, New York, NY 10451, USA; 4Amsterdam Public Health Research Institute, Amsterdam UMC, 1105 AZ Amsterdam, The Netherlands; mireille.van-poppel@uni-graz.at; 5Institute of Sport Science, University of Graz, 8010 Graz, Austria; 6Department of Epidemiology, University of Groningen, University Medical Center Groningen, 9700 RB Groningen, The Netherlands; h.groen01@umcg.nl; 7Department of Obstetrics and Gynecology, University of Groningen, University Medical Center Groningen, 9700 RB Groningen, The Netherlands; a.hoek@umcg.nl; 8Department of Obstetrics and Gynecology, Monash University, Monash Medical Center, Clayton, VIC 3800, Australia; ben.mol@monash.edu

**Keywords:** maternal obesity, childhood obesity, lifestyle intervention, cardiometabolic health, programming, follow-up

## Abstract

Maternal obesity is associated with adverse metabolic outcomes in her offspring, from the earliest stages of development leading to obesity and poorer cardiometabolic health in her offspring. We investigated whether an effective preconception lifestyle intervention in obese women affected cardiometabolic health of their offspring. We randomly allocated 577 infertile women with obesity to a 6-month lifestyle intervention, or to prompt infertility management. Of the 305 eligible children, despite intensive efforts, 17 in the intervention and 29 in the control group were available for follow-up at age 3–6 years. We compared the child’s Body Mass Index (BMI) Z score, waist and hip circumference, body-fat percentage, blood pressure Z scores, pulse wave velocity and serum lipids, glucose and insulin concentrations. Between the intervention and control groups, the mean (±SD) offspring BMI Z score (0.69 (±1.17) vs. 0.62 (±1.04)) and systolic and diastolic blood pressure Z scores (0.45 (±0.65) vs. 0.54 (±0.57); 0.91 (±0.66) vs. 0.96 (±0.57)) were similar, although elevated compared to the norm population. We also did not detect any differences between the groups in the other outcomes. In this study, we could not detect effects of a preconception lifestyle intervention in obese infertile women on the cardiometabolic health of their offspring. Low follow-up rates, perhaps due to the children’s age or the subject matter, combined with selection bias abating contrast in periconceptional weight between participating mothers, hampered the detection of potential effects. Future studies that account for these factors are needed to confirm whether a preconception lifestyle intervention may improve the cardiometabolic health of children of obese mothers.

## 1. Introduction

About 25% of children worldwide are overweight or obese [1,2]. Early life adiposity impairs cardiovascular and metabolic functioning during childhood and adolescence itself, and increases risks of cardiovascular disease (CVD) later in life [3]. There is emerging evidence that an elevated Body Mass Index (BMI) in mothers before conception and during pregnancy affects early embryonic development. Obese women have oocytes with altered mitochondrial function, leading to increased redox states, which are suggestive of oxidative stress in the zygote [4]. Large studies in assisted reproduction in women with obesity showed that using autologous embryos have lower success rates of liveborn than using donor embryos of normal weight women, suggesting obesity directly affects the embryo itself [5]. These early alterations in embryo development may increase the offspring’s risks of childhood obesity, increased fat mass, elevated blood pressure and disturbances in lipid and glucose concentrations. Additionally, this likely leads to increased adult cardiometabolic morbidity and all-cause mortality [6,7,8]. Hence, the World Health Organization (WHO) and a recent Lancet series have called to action to evaluate possible interventions to optimize preconception health [9,10]. The number of women of reproductive age with overweight and obesity is rising rapidly and is estimated to afflict more than a third of pregnancies [1,11]. Indicating the urgency to assess whether a preconception lifestyle change might be able to reduce the intergenerational development of obesity and improve the offspring’s long-term cardiometabolic health [12,13].

Animal studies have shown that improving diet in pregnancy has positive effects on the offspring’s adiposity and metabolic health [14]. Additionally, increasing preconception exercise improved the offspring’s adiposity and lipid levels [15]. Follow-ups of human randomized controlled trials (RCT) aimed to improve maternal lifestyle in women with obesity before pregnancy are currently lacking. Studies of lifestyle interventions during pregnancy have shown limited effects on offspring health; one study found improved infant skinfolds at 6 months of age but no change in the BMI [16], and another showed improved weight-for-age Z scores at 1 year without an effect on weight-for-length Z scores [17]. Six other studies found no effects on adiposity or other CVD risk factors at infancy up to 7 years of age [18]. Here, we present, for the first time, the effects of a preconception lifestyle intervention trial, which compared a 6-month lifestyle intervention program targeting physical activity, diet and behavior modification prior to infertility management to a control group receiving infertility management as usual [19]. The intervention in this trial indeed improved maternal lifestyle, through changes in diet and an increase in physical activity and resulted in an approximately 4-kg weight loss and halving the odds of metabolic syndrome after 6 months, and in those women who successfully lost weight, beneficial effects were seen up to 6 years later [20,21,22]. As beneficial intervention effects were found in all the women participating in the original RCT, we found it imperative to test our hypothesis that a healthier lifestyle before conception in infertile women with obesity improves the child’s cardiometabolic health at age 3–6 years.

## 2. Materials and Methods

### 2.1. Lifestyle Intervention

This follow-up study is based on the LIFEs*tyle* study (NTR1530), a multicenter RCT performed between 2011 and 2014 in 23 hospitals in the Netherlands [19]. Both the original study and this follow-up study were approved by the medical ethics committee of the University Medical Centre Groningen, the Netherlands (NL24478.042.08). Both parents gave written informed consent. Reporting in this manuscript adheres to the CONSORT 2010 guidelines.

The original randomized trial’s design has been described in detail previously [19]. In summary, infertile women (unsuccessfully tried to conceive for at least 12 months) with a BMI ≥ 29 kg/m^2^ were randomly allocated to a 6-month lifestyle intervention program preceding infertility management or to the control arm with prompt infertility management, both for a maximum of 24 months. Infertility management was individually assigned according to Dutch guidelines.

The intervention comprised of individualized motivational counseling by trained study nurses and dieticians to improve dietary intake and increase physical activity. Women were counseled to reduce their caloric intake by 600 kcal per day, and to adhere to Dutch guidelines for a balanced diet. Women wore a step-counter and aimed to achieve 10,000 steps a day and at least 30 min of moderate-intensity exercise two or three times per week. The intervention included six face-to-face and four phone consultations over 24 weeks and aimed at a weight loss of 5–10% of body weight or achieving a BMI < 29 kg/m^2^. The intervention stopped when conception occurred or was followed by up to 18 months of standard infertility management.

### 2.2. Eligibility for Follow-Up

The protocol of this follow-up study has previously been published [23]. In summary, all babies from a singleton pregnancy conceived within 24 months after their mother was randomized and who were known to be alive were eligible to participate in the follow-up assessments (*n* = 305 children of *n* = 577 randomized women) [19]. Children were 3–6 years old and living in the Netherlands (*n* = 300; *n* = 3 intervention group and *n* = 2 control group with no known address) when they were approached by mail (lay-man oriented information leaflets), and if possible, by repeated phone calls, for inclusion to the follow-up study. Parents were previously asked to fill out questionnaires about their child, during which parents could opt-out of being contacted by phone [23].

### 2.3. Follow-Up Assessment

Children were assessed in 2016 and 2017. We used a mobile research vehicle enabling us to conduct all assessments at/near the participant’s home. One parent was present during measurements, which were performed by two assessors from a pool of six experienced assessors with appropriate training. The assessors remained blinded to the lifestyle intervention allocation of the mother of the child undergoing assessment.

Maternal- and pregnancy-related characteristics were collected during the initial trial. Child’s birth weight was calculated as a gestational age and gender adjusted Z score based on Dutch reference curves with the LMS (lambda-mu-sigma) methodology [24,25].

At follow-up, we measured height using a SECA^®^ (Hamburg, Germany) 206 wall attached measuring tape to the nearest 0.1 cm. Weight was measured using a SECA^®^ 877 digital scale to the nearest 0.1 kg. We calculated BMI as weight in kg divided by height in meters squared. We calculated an age and sex adjusted Z-score based on the WHO reference values [26]. We measured waist and hip circumference with a SECA^®^ 201 measuring tape to the nearest 0.1 cm. We assessed body composition by bioelectrical impedance analysis (BIA) using the Bodystat 1500 mdd^®^ (Douglas, UK). Children were asked to empty their bladder and refrain from drinking at least 90 min prior to the BIA measurements. We used a validated, age-appropriate equation to calculate body-fat percentage (BF%) [27]. Measurements were taken in duplicate, in case there was considerable discrepancy between measurements, e.g., more than 0.5 kg for weight, a third measurement was obtained. All measurements were averaged.

We measured blood pressure (BP) seated in triplicate on the non-dominant arm, after 5 min of rest. A validated oscillometric device with age-appropriate cuff (Omron HBP-1300^®^, Kyoto, Japan) was used to measure BP [28]. We averaged the systolic blood pressure (SBP) and diastolic blood pressure (DBP), respectively, followed by a calculation of age and sex adjusted Z scores based on National Institute of Health (NIH) reference values [29]. We assessed arterial stiffness by Complior^®^ (ALAM Medical, St. Quentin Fallavier, France) to measure pulse wave velocity (PWV). This technique calculates the time between standardized measurement of the carotid and femoral pulses. By dividing the time by the distance between these two reference points, the PWV was calculated. 

With parental consent, a venous blood sample after an overnight fast was taken from their child during a separate appointment. We measured triglycerides (TG), total cholesterol (TC), high density lipoprotein (HDL) cholesterol, low density lipoprotein (LDL) cholesterol, insulin and glucose. We calculated the homeostatic model assessment of insulin resistance (HOMA-IR) as insulin (µIU/mL) times glucose (mmol/L) divided by 22.5.

### 2.4. Statistical Analyses

The sample size of the initial trial was set at 285 women per group based on the primary outcome (a healthy livebirth) [19]. No formal sample size calculation was performed for the current analysis, but the sample size of the initial trial was deemed sufficient to detect potentially relevant differences in the offspring, provided that participation rates were good.

To assess sample bias, we compared maternal- and pregnancy-related characteristics of the participating children to those that were eligible but did not participate. Similarly, of those children that participated, we compared maternal- and pregnancy-related characteristics from mothers in the intervention and control group. Differences in characteristics and outcomes were examined using Student’s *t*-test or Fisher’s exact test, as appropriate (Table 1).

As our primary analysis, we compared outcome measures (BMI Z score, BF%, BP Z score, PWV, serum lipids, glucose and insulin concentrations and HOMA-IR) of children from mothers in the intervention and control groups by means of Student’s *t*-test or Fisher’s exact test, as appropriate (Table 2). We adhered to the randomized trial design in our analyses. We tested in multivariable linear regression analyses whether an adjustment for the maternal- or pregnancy-related characteristics that were found to be different between group characteristics (i.e., birth weight), would alter the outcomes.

As exploratory analyses for potential offspring sex differences in effects of maternal obesity, we compared outcome measures for boys and girls separately (Appendix A). Furthermore, we assessed whether children of women who successfully lost weight (i.e., 5–10% weight reduction or achieving a BMI < 29 kg/m^2^) differed in outcomes from children of mothers who were unsuccessful, independent of randomization (Appendix A).

Values are presented as means and standard deviations (±SD) for continuous data and as frequency distributions for categorical data. We considered *p*-values of less than 0.05 statistically significant.

## 3. Results

Figure 1 shows the flowchart of the included participants. Out of the 163 women who conceived within 24 months in the invention group, 15 children were from twin/triplet pregnancies and three children were deceased, leaving 145 eligible children. From the 178 conceiving women in the control group, 14 children were from twin pairs and four children were deceased, leaving 160 children eligible. Despite intensive efforts, many parents did not respond or declined participation. A total of 51 parents provided informed consent; however, not all were able/willing to undergo measurements. Thus, we report on 17 children whose mother was randomized to the intervention group and 29 from the control group (15%).

As shown in Table 1, there were no differences between the maternal baseline characteristics and the pregnancy-related characteristics between the mothers of participating children and eligible children that did not participate. Additionally, the baseline and pregnancy-related characteristics of the mothers of participating children did not differ between the intervention and control groups. In the original trial, analyzing women who conceived and who did not conceive, the women in the intervention group had improved their lifestyle, lost weight and improved metabolic indices over 6 months [21]. Up to 6 years after the intervention, the differences abated between the intervention and control groups; however, beneficial effects were still present in the women who achieved the lifestyle intervention goals (5–10% weight loss or a BMI < 29 kg/m^2^) [22]. In our selected sample, all the women reduced their BMI slightly, but between the intervention and control groups there were no significant differences in the change in BMI between baseline and time of conception. Furthermore, maternal weight gain during pregnancy was equally high in both groups.

In the children, the mean birth weight of those who participated in the follow-up was lower in the intervention group compared to the control group (3234 g vs. 3652 g, *p* < 0.05; Table 1), while there was no statistically significant difference in the original trial (3312 g vs. 3341 g), and there were no differences in gestational age [19].

The mean (±SD) age of the participating children was 4.6 (±1.0) years (range 3.2–6.5 years). There were 22 boys (48%). Overall, the BMI Z score was 0.65 (±1.26), and the SBP and DBP Z scores were 0.51 (±0.59) and 0.94 (±0.60), respectively. Table 2 shows the cardiometabolic health indices of children in the intervention and control groups. We found no differences between the children of mothers from the intervention group compared to the children of mothers from the control group in childhood outcome measures (BMI Z score, BF%, BP Z score, PWV, serum lipids, glucose and insulin concentrations and HOMA-IR). Adjusting for confounders such as the child’s birth weight did not alter the effect estimates in multivariate analyses.

In exploratory analyses, cardiometabolic outcome values of boys and girls did not differ according to the maternal allocation to lifestyle intervention (Appendix A). Furthermore, there were no differences in childhood cardiometabolic outcomes between children of mothers who successfully lost weight (*n* = 8) compared to those whose mothers did not (Appendix A). The latter analysis was independent of the assigned groups by randomization. In both exploratory analyses, the groups had very small numbers prohibiting conclusions to be drawn, and too few lab values were available for the children of mothers who successfully lost weight.

## 4. Discussion

We could not detect differences in offspring cardiometabolic health at age 3–6 years in the follow-up of a preconception maternal lifestyle intervention trial. Despite numerous efforts to enhance participation (e.g., measurements near the participant’s home, multiple phone calls, layman information leaflets), our study was hampered by high attrition rates, which reduced the statistical power substantially. In our sample, the maternal BMI at conception and gestational weight gain (GWG) throughout pregnancy were not different between the intervention and control groups. We were unable to examine maternal cardiovascular and metabolic factors at the time of conception in our selected sample. However, we do know from previous publications by our group, that in all the women in the original trial, regardless of their conception status, the intervention increased physical activity, reduced snacking and sugary drinks intake, led to weight loss (approximately 4 kg) and halved the odds of metabolic syndrome after 6 months [20,21]. Furthermore, up to six years later, the intervention led to decreased caloric intake, and those women that were deemed successful in the primary trial showed improved BMI and cardiometabolic indices [20,22].

A range of experimental animal studies and observational human studies have shown that during embryonic developmental, even small environmental changes will have lasting effects (Table 3) [13]. In animals, changing the maternal lifestyle before and during pregnancy, and, in turn, comparing between in utero exposure to maternal obesity or reduced weight, was associated with improved adiposity and lipid levels in offspring [14,15]. In humans, environmental factors and the lifestyle of mothers impacted the fetal and placental metabolism, oxidative stress and interactions of these, inflicting epigenetic changes that are suggested to have lasting effects [30,31]. Furthermore, in assisted reproduction variations in embryo culture conditions have led to altered metabolic and epigenetic regulation, resulting in altered growth and cardiometabolic profiles of offspring [32,33]. In a large cohort study, children born after assisted reproduction had different growth patterns in their first few years, but ended up at a grossly similar height and weight in adolescence compared to their naturally conceived peers [34]. Although these findings seem reassuring, such alterations in growth during early life are linked to a predisposition of poor cardiometabolic health later in life, suggested by early life echocardiograms alterations in cardiac shape and function in assisted reproduction offspring [35].

This follow-up was based on the first randomized controlled trial in obese women examining the effects of a preconception lifestyle intervention. Due to the randomized design, confounding factors related to maternal infertility and/or obesity were equally divided between groups. Hence, we consider this population of infertile women valid to explore the effects of maternal preconception lifestyle change and weight loss on offspring. Despite the fact that most maternal- and pregnancy-related factors were similar between the groups, there was selective participation in our follow-up sample. This was indicated by a lower birth weight in children in the intervention group, a difference that was not present in the original trial [19]. Since our study had a null result and adjusting our analyses for birth weight did not change our results, we consider it unlikely that this selection has led to bias.

More importantly, in contrast to the original trial, participating women from both groups had similar weight loss between randomization and periconception. Additionally, the GWG was above the recommended levels for obese women (5–9 kg) in both groups [46]. This resulted in a very limited contrast in maternal body weight at time of conception between the groups, which may have contributed to our null finding. While the intervention induced effects on maternal parameters other than body weight, i.e., glucose metabolism or nutritional quality [20,22], these changes may not have been able to mitigate the detrimental effects of maternal obesity and/or excessive GWG on the offspring’s cardiometabolic health [47]. Since studies assessing children born after their mothers had bariatric surgery showed improved cardiometabolic health [48,49], more substantial maternal weight loss may be needed to elicit changes in childhood health outcomes. On the other hand, observational evidence suggests a graded ‘dose response’ association between maternal BMI and offspring’s cardiometabolic health [47,50], indicating even modest weight changes could carry positive effects, but we were not able to detect any effect.

To provide more reliable conclusions about the potential effects of maternal lifestyle change before conception on children, future studies should aim to maximize follow-up rates and power calculations based on childhood outcomes, which should account for high attrition. Compared to our study, a higher participation rate of 52% was present 3 years after a different lifestyle intervention during pregnancy [51]. Still, attrition in follow-up studies is generally high [52], and reasons why often remain unknown. Although parents were not required to state their reasons for declining participation, those that did indicated that they refused due to time restraints, did not want to burden their child or that they mainly wanted to become pregnant and were not interested in a further follow-up. Some stated that they were aware of the negative association of maternal obesity with childhood health, while many others were not previously aware; many women indicated that they did not want to contribute to further evidence of that association. While we attempted to involve participants in the planning of the study, only a few provided limited information on topics of interest to them that corresponded to the outcomes we were considering. The future investigation of offspring health-related themes considered relevant by obese women and their partners may provide guidance into a strategy that achieves lower attrition rates in follow-up studies such as our own.

Individual interventions in obese adults have been marginally successful [52]; community-based interventions and policies could thus be better suited to optimize the health of women prior to pregnancy. While the WHO has made an important step in global obesity prevention by formulating nine voluntary targets to prevent non-communicable diseases [53], and policy change has shown some local improvements [54], these have not yet been able to counteract the overall worldwide burden of obesity. Since pregnant women with obesity indicated they were mostly unaware of the effects of obesity on their (future) child [55], there is an urgent need to improve awareness in the general public of the consequences of obesity before and during pregnancy. As health behaviors are not solely individually determined and depend on environmental factors [56], and knowledge of healthy habits do not directly translate to changes in behavior [57], policies focusing on improving nutrition, physical activity and sports involving the home, school/work environment and the community are needed to curb the intergenerational cycle of obesity.

## 5. Conclusions

We could not detect any effect of a preconception lifestyle intervention—that did improve lifestyle, induced weight loss and improved cardiometabolic health in women 6 months after randomization—on the offspring’s cardiometabolic health at age 3–6 years. Our study was hampered by limited statistical power, perhaps due to the children’s age or the subject matter, as well as the minimal difference in the maternal periconception weight between the groups of participants in contrast to the original trial. Future studies should account for these factors to maximize follow-up rates to be able to draw conclusions about the potential of preconception lifestyle interventions to affect offspring cardiometabolic health. Whether there is no effect of lifestyle interventions in women with obesity prior to conception on their offspring’s cardiometabolic health needs to be confirmed in larger studies.

## Figures and Tables

**Figure 1 cells-11-00041-f001:**
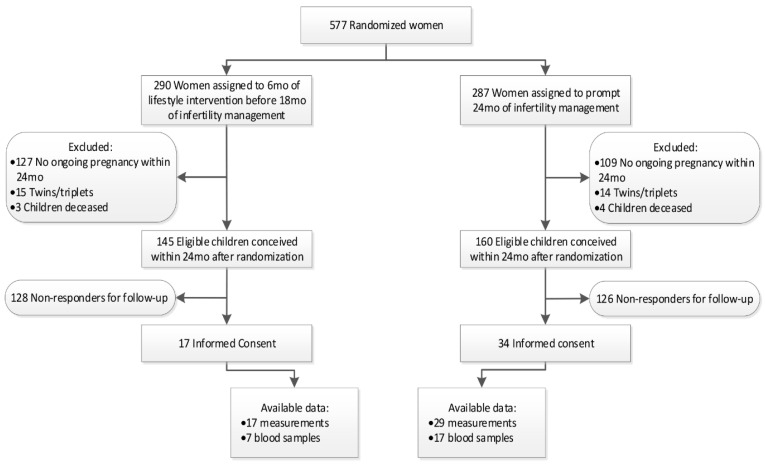
Flowchart of included participants. Mo = Months.

**Table 1 cells-11-00041-t001:** Maternal baseline and pregnancy related characteristics.

	*n*	Intervention	*n*	Control	*n*	Non-Participants	*p*-Value ^#^
Maternal baseline characteristics:							
Age, years—mean (SD)	17	29.9 (3.4)	29	29.3 (4.1)	259	29.1 (4.3)	0.51
Caucasian—no (%)	17	16 (94.1)	29	28 (96.6)	259	227 (87.6)	0.13
Education—no (%)	17	-	29	-	246	-	0.36
Primary school	-	0 (0.0)	-	0 (0.0)	-	10 (4.1)	-
Secondary education	-	2 (11.8)	-	8 (27.6)	-	59 (24.0)	-
Intermediate vocation education	-	11 (64.7)	-	15 (51.7)	-	115 (46.7)	-
Higher education	-	2 (13.3)	-	6 (20.7)	-	62 (25.2)	-
Smoker—no. (%)	17	3 (17.6)	29	5 (17.2)	255	56 (22.0)	0.56
BMI (kg/m^2^)—mean (SD)	17	36.0 (2.7)	29	35.6 (3.0)	259	35.9 (3.5)	0.80
Pregnancy related characteristics:							
Maternal age at time of pregnancy (years)—mean (SD)	17	30.5 (3.4)	29	29.8 (4.3)	254	29.8 (4.4)	0.69
Nulliparous—no. (%)	17	13 (76.5)	29	21 (72.4)	258	207 (80.2)	0.43
Delta baseline BMI and periconceptional BMI—mean (SD)	15	−0.7 (2.8)	24	−0.9 (1.5)	103	−1.0 (2.7)	0.24
Gestational weight gain (kg)—mean (SD)	10	11.8 (6.1)	18	11.3 (5.8)	195	9.9 (6.3)	0.64
Gestational diabetes—no. (%)	17	4 (23.5)	29	7 (24.1)	252	44 (17.5)	0.31
Gestational age at birth (weeks)—mean (SD)	17	39.0 (1.7)	29	39.2 (1.7)	254	39.0 (2.1)	0.79
Birth weight (grams)—mean (SD)	17	3234 (497) *	29	3652 (454) *	253	3391 (585)	0.25
Conception mode—no (%)	17	-	29	-	255	-	0.78
Natural	-	10 (58.8)	-	8 (27.6)	-	97 (38.0)	-
Ovulation Induction	-	5 (29.4)	-	11 (37.9)	-	78 (30.6)	-
IUI	-	2 (11.8)	-	5 (17.2)	-	37 (14.5)	-
IVF/ICSI/CRYO	-	0 (0.0)	-	5 (17.2)	-	43 (16.9)	-
Breastfeeding ^+^—no (%)	17	4 (23.5)	29	9 (31.0)	259	70 (27.0)	0.86

^#^ Comparison between participants versus non-participants. * *p* < 0.05 between intervention and control. ^+^ Exclusive breastfeeding for three or more months. BMI = Body mass index, kg = kilogram, IUI = Intra-uterine insemination, IVF = In Vitro fertilization, ICSI = Intracytoplasmic sperm injection, CRYO = Cryotherapy.

**Table 2 cells-11-00041-t002:** Cardiometabolic outcome values of children of mothers from the intervention and control group.

	Anthropometry				
	*n*	Intervention	*n*	Control	95% CI
BMI (Z-score)— mean (SD)	16	0.69 (1.17)	28	0.62 (1.04)	−0.62–0.76
Waist circumference (cm)— mean (SD)	17	53.4 (4.3)	29	53.4 (5.3)	−3.04–3.10
Hip circumference (cm)— mean (SD)	17	58.3 (4.4)	29	58.4 (6.9)	−3.90–3.66
Body-fat (%)— mean (SD)	16	20.7 (7.8)	26	21.2 (9.4)	−6.16–5.16
	Cardiovascular				
	*n*	Intervention	*n*	Control	95% CI
SBP (Z-score)— mean (SD)	16	0.46 (0.65)	27	0.54 (0.57)	−0.46–0.30
DBP (Z-score)— mean (SD)	16	0.91 (0.66)	27	0.96 (0.57)	−0.44–0.33
PWV (m/sec)— mean (SD)	12	4.51 (0.83)	22	4.50 (1.14)	−0.75–0.77
	Metabolic				
	*n*	Intervention	*n*	Control	95% CI
Triglycerides (mmol/L)— mean (SD)	7	0.71 (0.63)	17	0.53 (0.17)	−0.39–0.76
Total cholesterol (mmol/L)— mean (SD)	7	4.26 (0.79)	17	4.07 (0.54)	−0.39–0.77
LDL cholesterol (mmol/L)— mean (SD)	7	2.46 (0.65)	17	2.36 (0.40)	−0.35–0.54
HDL cholesterol (mmol/L)— mean (SD)	7	1.48 (0.20)	17	1.48 (0.26)	−0.22–0.24
Insulin (µIU/mL)— mean (SD)	7	5.52 (3.12)	12	4.21 (2.87)	−1.66–4.29
Glucose (mmol/L)— mean (SD)	7	4.70 (0.33)	17	4.47 (0.42)	−0.13–0.60
HOMA-IR— mean (SD)	7	1.19 (0.75)	12	0.87 (0.64)	−0.37–1.00

BMI = Body mass index, SBP = Systolic blood pressure, DBP = Diastolic blood pressure, LDL = Low-density lipoprotein, HDL = High-density lipoprotein, HOMA-IR = Homeostatic model of insulin resistance.

**Table 3 cells-11-00041-t003:** Summary of selected current (pre-)pregnancy lifestyle intervention studies in animal and human settings and effects on offspring’s health.

Study Identifier	Animal (A) or Human (H)	Intervention	Results
Gallou-Kabani et al., (2007) [36]	A	Dietary at time of conception/pregnancy and lacation	Female, not male offspring, had a higher proportion that remained lean on postnatal high fat diet and improved glycemic indices and lipids.
Zambrano et al., (2010) [14]	A	Dietary (30 days prior pregancy)	(Partial) normalization of fat mass, triglycerides, leptin and insulin
Dennison et al., (2013) [37]	A	Dietary (low fat high fiber) and/or sitagliptin (8 weeks prior pregnancy)	No changes in offspring body weight. Diet had no significant effects on energy intake, leptin, fasting glucose, however some microbiome change were seen. Sitagliptin alone had largest reduction in glucemic control.
Vega et al., (2015) [15]	A	Exercise (30 days prior pregnancy)	Reduced leptin, triglycerides, glucose
Xu et al., (2018) [38]	A	Dietary (up to 9 weeks prior preganncy)	Longer maternal diet intervention showed normalization of offspring’s glucose and lipid metabolism
Mustilla et al., (2012) [39]	H	Lifestyle intervention on diet and physical activity durng pregnancy	At 24–48 months, the offpsring in the intervention group had slower gains in BMI z score. Over the 0–48 months there was no differences in BMI z score gain between groups. (Follow-up rate, 72%)
Tanvig et al., (2014) [40]	H	Diet, exercise and coaching during pregnancy (RCT)	At 2.8 years follow-up, there were no differences between groups in BMI z-scores, nor in skinfold, anthropometrics, total fat mass, lean mass or fat percentage. (Follow-up rate 29%)
Rauh et al., (2015) [41]	H	Lifestyle intervention including dietary and physical acivity counseling twice during pregnancy	At 10–12 months after birth, there were no significant differences in offspring’s weight. (Follow-up rate, 85%)
Horan et al., (2016) [42]	H	Dietary intervention during pregnancy in women with previous LGA infant	No effects on offspring at 6 months, at 2 years improvement of anthropometrics indices with heatlhier dietary intake during pregancy. (Follow-up rate, 35%)
Kolu et al., (2016) [43]	H	Lifestyle intervention of diet and physical activity during 5 antenatel visits during pregnancy	No differences in child’s BMI up to 7 years. Children of mothers who adhered to all lifestyle aims had signifcantly lower BMIs. (Follow-up rate, 43%)
Vesco et al., (2016) [17]	H	Weekly weight management intervention focused on diet and exercise during pregnancy	At 1 year of age there was significant reduction in weight-for-age z scores in children in the intervention group, but no differences in weight-for-height z score between groups.
Ronnberg et al., (2017) [44]	H	Lifestyle intervention on diet and physical activity during pregnancy, focus on healthy gestational weight gain	Follow-up of children’s BMI until 5 years of age showed no differences between groups in child’s BMI z score. (Follow-up rate, 80%)
Dalrymple et al., (2021) [45]	H	Diet and physical activity intervention of 8 weeks during pregnancy (RCT)	6 months lower skinfold measures in interventions, at 3 year follow-up no significant differences in BMI or skinfold between groups. Significan lower pulse rate in offspring of intervention (Follow-up rate, 33%)

## Data Availability

The datasets used and/or analyzed during the current study are available from the corresponding author on reasonable request. This manuscript has been presented as a preprint in the University of Amsterdam repository for Ph.D. thesis.

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
