# Peer review of "The Effects of a Preconception Lifestyle Intervention on Childhood Cardiometabolic Health—Follow-Up of a Randomized Controlled Trial"

_cells, 2021, doi:10.3390/cells11010041_

Round 1
Reviewer 1 Report
The authors describe, that they did not find any effect of preconception lifestyle intervention. In the discussion the authors describe other animal and human studies which were able to show effects from maternal lifestyle change. I recommend to pool their data with data from other studies to improve case numbers and to overcome the limitations described in the conclusions section.
Author Response
Thank you for your time to review our manuscript, we have send a letter to the editor for further review.

Reviewer 2 Report
The authors have addressed all my previous comments sufficiently. I have no more comments
Author Response
Thank you for your time to review our manuscript and your comments.
This manuscript is a resubmission of an earlier submission. The following is a list of the peer review reports and author responses from that submission.
Round 1
Reviewer 1 Report
The authors describe that the intervention should inrease physical activity (lines 86-88) with a target BMI <29kg/m². But the data (table1) shows no difference between intervention (36kg/m²) and control (35.6kg/m²). Also the Non-participants group is far away from the target 29kg/m² . So I would say, that there is a lack in the design of the study and it makes no sense to analyze the childrens data further.
Also the discussion lacks of clarity: in line 207-209 the authors state "In women, the intervention increased physical activity, reduced snacking and sugary drinks intake, led to weight loss (approximately 4kg)...". This weight loss is not documented and the presented BMI data does not support weight loss.
I can not recommend this paper. Sorry for that.
Reviewer 2 Report
The whole paper is well written. The introduction provides a good, generalized background. The topic of the article is very interesting and potentially important for human health and disease intervention. It will be very interesting to see the results from a larger group, I hope that the authors will continue to explore this topic.
My main concern is how the groups were selected. The study aimed to investigate if lifestyle intervention can improve childhood metabolic health. But in the intervention group, we can find both women with changed and unchanged BMI. If there was no change in BMI how authors knew that lifestyle changes were implemented by these women? If nothing changes how this can be the experimental group? And how comparing this heterogeneous group with the control can produce informative results? Consider removing some data from the analysis or diving intervention group into two subgroups
My specific comments are listed below:
- Do authors check in any way if women enrolled in the control group do not change their BMI in the course of the last 6 months before the study?
- Figure 1: please consider to please consider moving figure 1 before the tables, closer to the first paragraph
- Table 1: please consider putting table 1 on one page, it will be easier to read
- Do authors have any cardiovascular or metabolic traits from women from the control and intervention group (especially before and after intervention)? Maybe not BMI itself but improvement of cardiovascular/lipid/metabolic factors have the effects on children health
- Supplementary table 2: Do authors have any metabolic traits for these groups?
- Supplementary table 2: the number of the sample considered in the successful group is 6-8 and 28-38 samples for the unsuccessful group but in figure 1 authors stated that for the intervention group they have only 17 available samples. Does it mean that in the supplementary table data from intervention and control has been pooled? If yes it should be clearly stated in the text or in the description of the table
Reviewer 3 Report
Thank you very much for the interesting study. It is a very important topic, but I have some major comments that should be addressed before publication.
In general, many things are assumed to be known, which only become clear to me as a reader when I also read the previous papers. Central here is the design of the intervention. So it remains completely open what kind of movement was offered and in what form. How much was implemented. How many were reached in general? This remains unclear in the abstract, but also in the entire text. Thus, the first large number of the total cohort is massively misleading.
The n-numbers in the supplement in the IG, e.g. separated by gender, are very small (e.g. n=3). The hip circumference is actually superfluous.
The data of the regression analysis and the underlying model are missing. Thus, relevant influencing factors, if any, are not presented. Overall, the methodology is not clearly presented and the results are presented too superficially.
In general, the authors are very self-critical of the data in the discussion; but overall, more questions are raised than answered by the study. As a result, the conclusion also goes in an incorrect direction. The fact that the authors do not describe any effect of lifestyle is not surprising to me given the way the study was done. The argument of the lack of statistical power is justified, but all in all, no scientific added value is recognisable in the present form.